# MATE1 Deficiency Exacerbates Dofetilide-Induced Proarrhythmia

**DOI:** 10.3390/ijms23158607

**Published:** 2022-08-03

**Authors:** Muhammad Erfan Uddin, Eric D. Eisenmann, Yang Li, Kevin M. Huang, Dominique A. Garrison, Zahra Talebi, Alice A. Gibson, Yan Jin, Mahesh Nepal, Ingrid M. Bonilla, Qiang Fu, Xinxin Sun, Alec Millar, Mikhail Tarasov, Christopher E. Jay, Xiaoming Cui, Heidi J. Einolf, Ryan M. Pelis, Sakima A. Smith, Przemysław B. Radwański, Douglas H. Sweet, Jörg König, Martin F. Fromm, Cynthia A. Carnes, Shuiying Hu, Alex Sparreboom

**Affiliations:** 1Division of Pharmaceutics and Pharmacology, College of Pharmacy and Comprehensive Cancer Center, The Ohio State University, Columbus, OH 43210, USA; uddin.33@osu.edu (M.E.U.); eisenmann.11@buckeyemail.osu.edu (E.D.E.); li.10991@osu.edu (Y.L.); huang.2834@buckeyemail.osu.edu (K.M.H.); garrison.220@buckeyemail.osu.edu (D.A.G.); talebi.9@osu.edu (Z.T.); gibson.972@osu.edu (A.A.G.); jin.1134@osu.edu (Y.J.); nepal.17@osu.edu (M.N.); pkpdfu@gmail.com (Q.F.); xinxin224@gmail.com (X.S.); 2Department of Physiology and Cell Biology, College of Medicine, The Ohio State University, Columbus, OH 43210, USA; ingrid.bonilla4@upr.edu; 3Division of Outcomes and Translational Sciences, College of Pharmacy, The Ohio State University, Columbus, OH 43210, USA; alec.miller@midwestern.edu (A.M.); tarasov.4@osu.edu (M.T.); radwanski.2@osu.edu (P.B.R.); carnes.4@osu.edu (C.A.C.); hu.1333@osu.edu (S.H.); 4Department of Pharmaceutics, School of Pharmacy, Virginia Commonwealth University, Richmond, VA 23298, USA; jayce@mymail.vcu.edu (C.E.J.); dsweet@vcu.edu (D.H.S.); 5Novartis Institute for Biomedical Research, East Hanover, NJ 07936, USA; xiaoming.cui@novartis.com (X.C.); heidi.einolf@novartis.com (H.J.E.); ryan.pelis@novartis.com (R.M.P.); 6OSU Wexner Medical Center, Department of Internal Medicine, Division of Cardiovascular Medicine, The Ohio State University, Columbus, OH 43210, USA; sakima.smith@osumc.edu; 7Dorothy M. Davis Heart and Lung Research Institute, The Ohio State University, Columbus, OH 43210, USA; 8Institute of Experimental and Clinical Pharmacology and Toxicology, Friedrich-Alexander-Universität Erlangen-Nürnberg, 91054 Erlangen, Germany; joerg.koenig@fau.de (J.K.); martin.fromm@fau.de (M.F.F.); 9Division of Pharmacy Practice and Science, College of Pharmacy, The Ohio State University, Columbus, OH 43210, USA

**Keywords:** dofetilide, organic cation transporters, arrhythmia, PBPK modeling

## Abstract

Dofetilide is a rapid delayed rectifier potassium current inhibitor widely used to prevent the recurrence of atrial fibrillation and flutter. The clinical use of this drug is associated with increases in QTc interval, which predispose patients to ventricular cardiac arrhythmias. The mechanisms involved in the disposition of dofetilide, including its movement in and out of cardiomyocytes, remain unknown. Using a xenobiotic transporter screen, we identified MATE1 (*SLC47A1*) as a transporter of dofetilide and found that genetic knockout or pharmacological inhibition of MATE1 in mice was associated with enhanced retention of dofetilide in cardiomyocytes and increased QTc prolongation. The urinary excretion of dofetilide was also dependent on the MATE1 genotype, and we found that this transport mechanism provides a mechanistic basis for previously recorded drug-drug interactions of dofetilide with various contraindicated drugs, including bictegravir, cimetidine, ketoconazole, and verapamil. The translational significance of these observations was examined with a physiologically-based pharmacokinetic model that adequately predicted the drug-drug interaction liabilities in humans. These findings support the thesis that MATE1 serves a conserved cardioprotective role by restricting excessive cellular accumulation and warrant caution against the concurrent administration of potent MATE1 inhibitors and cardiotoxic substrates with a narrow therapeutic window.

## 1. Introduction

Atrial fibrillation (AFib) is the most commonly encountered sustained cardiac arrhythmia in clinical practice globally [1], and it has been estimated that up to 6.1 million people in the United States alone suffered from atrial fibrillation between 1990 and 2013 [2]. The prevalence of atrial fibrillation is predicted to affect about 12 million people in the United States and 17 million in Europe by 2050, and people over 65 years of age are particularly prone to experience this condition [3,4]. Predictive models and preventative interventions have been proposed to identify patients at risk of AFib based on age, race, blood pressure, smoking, diabetes, and prior history of myocardial infarction, but primary prevention strategies for AFib have yet to be established [5,6]. Dofetilide, a class III antiarrhythmic drug, is widely used to prevent the recurrence of AFib and flutter by acting on a rapid delayed rectifier potassium current (*I_kr_*) and inhibition of the efflux of potassium [7]. Dofetilide does not cause any negative inotropic, dromotropic, and chronotropic effects observed with other commonly prescribed antiarrhythmic drugs, such as amiodarone, dronedarone, or sotalol [8]. Furthermore, due to the lack of serious extra-cardiac adverse effects and relatively low morbidity rate as compared with other antiarrhythmic drugs, dofetilide is commonly used in the treatment of AFib. However, the clinical use of dofetilide is associated with extensive inter-individual pharmacokinetic variability, and mechanisms underlying this variability remain unclear. This variability has clinical implications, since elevated plasma concentrations of dofetilide have been associated with QTc prolongation and increase the risk of *torsade de pointes*, a potentially lethal cardiac arrhythmia [9].

It has been reported that females are exposed to higher dofetilide plasma concentrations (14–22%) than males [10] and the rate of discontinuation and dose reduction due to QTc prolongation is higher in females [11]. Dofetilide undergoes extensive renal tubular secretion [10], and the initial dofetilide dose is adjusted based on an individual’s renal function, as estimated from the clearance of creatinine. Despite the presence of a narrow therapeutic index that requires in-hospital initiation of therapy, mechanistic details of dofetilide elimination and the underlying causes of recurrent or drug-induced arrhythmias remain poorly understood [12]. Although previous studies have reported the involvement of several transporters facilitating the transport of antiarrhythmic drugs [13,14], and potentiating the blockage of the hERG potassium channel [15], the mechanisms involved in the movement of dofetilide in and out of cells remain unknown. We set out to remedy this knowledge deficit by employing genetic and pharmacological strategies that allow the identification of solute carriers capable of transporting dofetilide in a manner that would influence susceptibility to toxicity. This analysis resulted in the identification of the multidrug and toxin extrusion protein MATE1 (*SLC47A1*), an organic cation transporter predominantly expressed in the kidney and heart [16,17,18], as a previously unrecognized transporter of dofetilide in cardiomyocytes and renal tubular cells. In addition, we found that dofetilide transport by MATE1 is highly sensitive to pharmacological inhibition, in line with its known role in the distribution and elimination of endogenous and exogenous substrates [16,19,20,21,22,23], and suggest that these findings have ramifications for optimizing polypharmacy regimens in subjects requiring treatment with cardiotoxic MATE1 substrates.

## 2. Results and Discussion

### 2.1. Identification of MATE1 as a High-Affinity Carrier of Dofetilide

Dofetilide is a basic compound that is partially ionized under normal physiological conditions (pH 7.4) [24], suggesting that its transcellular movement is dependent on a transport mechanism mediated by one or more members of the class of solute carriers. To identify transporters of interest for further consideration, we initially performed a screen in HEK293 or CHO cells engineered to overexpress the human organic cation transporters OCT1 (*SLC22A1*), OCT2 (*SLC22A2*), OCT3 (*SLC22A3*) or OCTN1 (*SLC22A4*), the organic anion transporters OAT1 (*SLC22A5*) or OAT3 (*SLC22A7*), and the multidrug and toxin extrusion proteins MATE1 (*SLC47A1*) or MATE2-K (*SLC47A2*), which are known to be involved in the transport of a broad range of xenobiotic substances. The results of this screen indicated that dofetilide was most efficiently transported by MATE1 (4.3-fold increased uptake compared to control cells; Figure 1A). Although the physiological role of MATE1 supports luminal efflux of organic cations, the kinetics of MATE1-mediated transport in screens such as the one we performed characterizes the interaction of substrates with the outward-facing transporter by measuring uptake rather than efflux. The transport of dofetilide and the positive control substrate tetraethylammonium (TEA) by MATE1 in our engineered mammalian cells was further found to be sensitive to pharmacological inhibition (Figure 1B), time-dependent (Figure 1C), and saturable with a Michaelis-Menten constant of 6.72 ± 1.71 µM and a maximum velocity of 544 ± 55.7 pmol/min/mg, respectively (Figure 1D). Similar observations were made in cells overexpressing mouse MATE1 (Appendix A), the single orthologous murine transporter of MATE1 and MATE2-K in humans [20]. In support of the interaction of dofetilide with MATE1, we found that dofetilide was able to dose-dependently inhibit the MATE1-dependent transport of various known substrates, including TEA and metformin (Appendix A).

Since MATE1 is the most highly expressed organic cation transporter in the mammalian heart [16,18], we hypothesized that genetic deficiency or pharmacologic inhibition of MATE1 could lead to altered retention of dofetilide in cardiomyocytes and modulate its downstream effects. To test this hypothesis, we examined the ex vivo accumulation of dofetilide in cardiomyocytes isolated from wild-type mice and MATE1^−/−^ mice, both on an FVB background, using a Langendorff perfusion system [25,26]. We confirmed that MATE1 is expressed in the heart and kidneys of wild-type mice (Figure 1E and Appendix A), an observation that is consistent with previously reported studies [22,27] showing the presence of MATE1 in the murine heart and the absence of any intrinsically abnormal pathological or biochemical changes in MATE1^−/−^ mice [28]. The ex vivo studies indicated that the accumulation of dofetilide in cardiomyocytes is time dependent and significantly increased in the absence of MATE1 (Figure 1F). Interestingly, pretreatment of cardiomyocytes with the MATE1 inhibitor cimetidine was associated with a substantially diminished accumulation of dofetilide in cells from both wild-type mice and MATE1^−/−^ mice (Figure 1G). This finding is consistent with the thesis that the initial uptake of dofetilide into cardiomyocytes occurs via a mechanism that is sensitive to cimetidine-mediated inhibition.

In line with the ex vivo cardiomyocyte data, we found that MATE1-deficiency was associated with significantly increased in vivo accumulation of dofetilide in heart tissue (Figure 1H and Appendix A). Similar observations were made with the MATE1 substrate TEA, and this phenotype was sensitive to pretreatment with cimetidine (Figure 1I). To provide further support for the thesis that an analogous transport system is operational in humans, we confirmed that the accumulation of dofetilide and TEA in human cardiomyocytes was also sensitive to pretreatment with cimetidine (Figure 1J). These findings further support our hypothesis that dofetilide is a transported substrate of MATE1, and that genetic deficiency or pharmacological inhibition of MATE1 alters the cardiac accumulation of dofetilide.

### 2.2. MATE1 Deficiency Exacerbates Dofetilide-Induced Proarrhythmia

Since concentrations of dofetilide are positively correlated with drug-induced QTc prolongation [29,30], we next tested the hypothesis that inhibition of MATE1 triggers the onset of QTc prolongation following treatment with dofetilide. Because dofetilide prolongs QTc intervals in neonates but not in adult mice, neonatal mice were used in these studies [31], and we verified that MATE1 is expressed in the neonatal heart and kidney of wild-type mice (Figure 2A). Following a single i.p. injection of dofetilide, a significant prolongation in QTc interval was observed in neonatal MATE1^−/−^ mice (Figure 2B,C) and ECG analysis indicated that these animals developed second-degree (Mobitz I and II) atrioventricular blocks (Figure 2D–F), suggesting that MATE1 acts indirectly as a modulator of the *I*_Kr_ blockade. Although results from these experiments provide proof-of-principle that inhibition of the cardiac efflux transporter MATE1 exacerbates the *I_Kr_* blockade, further investigation is required to identify the cimetidine-sensitive mechanism by which dofetilide is taken up into cardiomyocytes (Figure 2G). Regardless of the identity of this mechanism, the scenario proposed for dofetilide is congruent with previously reported cardiac transport of other cardiovascular drugs such as verapamil, which is taken up by OCTN2 (*SLC22A5*) and extruded by ABCB1 (P-gp) [14], and quinidine, for which uptake by OCTN1 is a prerequisite for its ability to induce hERG channel blockade [15].

### 2.3. Inhibition of MATE1 Attenuates Renal Elimination of Dofetilide

Previous clinical studies have indicated that the inhibition of the renal cation secretory pathway by agents such as cimetidine and verapamil can increase the plasma concentration of dofetilide by 53–93% [32,33]. In order to evaluate the contribution of MATE1 to these drug-drug interactions and to the renal transport of dofetilide in a physiologically-relevant model system, we performed basolateral to apical flux studies in Madin-Darby canine kidney (MDCK) epithelial cells [34,35], which differentiate to form a polarized monolayer [36]. We found that in this model the vectorial basolateral-to-apical flux of metformin, a known substrate of OCT2 and MATE1 [37], was dependent on the expression of both OCT2 on the basolateral membrane and MATE1 on the apical membrane (Figure 3A). Unexpectedly, the transepithelial flux of dofetilide across MDCK monolayers was also dependent on the presence of both OCT2 and MATE1 (Figure 3B). To reconcile the paradoxical observation that overexpression of OCT2 in HEK293 cells did not facilitate the uptake of dofetilide (Figure 1A), it is worth pointing out that precedent studies with certain organic cations in non-polarized cells have revealed false negative results compared to experiments performed using polarized monolayers [38].

Based on the in vitro findings in MDCK cells, we hypothesized that the inhibition of OCT2 and/or MATE1 in vivo could potentially lead to decreased urinary excretion and a concomitant increase in the dofetilide concentration in plasma. In a mass balance study, we found that deficiency of either MATE1 or both OCT1/OCT2 (OCT1/2), the murine orthologues of human OCT2, as well as MATE1 significantly reduced the urinary excretion of dofetilide compared with wild-type mice or OCT1/2^−/−^ mice (Figure 3C), regardless of sex (Appendix A). To examine the implications of altered renal secretion on the plasma concentrations of dofetilide, we performed pharmacokinetic analyses after the oral (5 mg/kg) (Figure 3D–F) and i.v. (2.5 mg/kg) (Appendix A) administration of dofetilide. The results of these studies revealed modest increases in the plasma concentrations of dofetilide in OCT1/2- or MATE1-deficient mice, while deficiency of all three transporters was associated with more pronounced elevations in measures of systemic exposure (Figure 3D–F). These observations suggest that there might be shunting of dofetilide to alternative routes of elimination in the absence of MATE1, that alternate distribution profiles compensate for MATE1 loss in the kidney such that measures of systemic exposure remain unchanged, and that additional basolateral transporters exist that regulate the movement of dofetilide from the circulation into tubular cells in the absence of OCT1 and OCT2. Most importantly, these observations imply a simultaneous dependence of OCT1/2 and MATE1 impairment on circulating dofetilide concentrations [16,17], a thesis that led us to further explore the mechanistic basis of known and unknown pharmacokinetic drug-drug interactions with dofetilide.

### 2.4. Drugs Contraindicated for Use with Dofetilide Inhibit MATE1 Function

Previous studies demonstrated that several drugs are contraindicated in subjects receiving treatment with dofetilide due to drug-drug interactions that affect hepatic metabolism or renal excretion [24,39,40,41]. Interestingly, the contraindicated drugs cimetidine, dolutegravir, megestrol, and prochlorperazine are known to interfere with an unidentified renal cation transport system and elevate the plasma concentrations of several xenobiotic organic cations [42,43,44]. Based on this prior knowledge, we hypothesized that these agents might also inhibit the MATE1-mediated urinary excretion of dofetilide. Indeed, we found that many of the contraindicated agents, including cimetidine, verapamil, vandetanib, megestrol, trimethoprim, ketoconazole, and itraconazole, potently inhibit the MATE1-dependent transport of dofetilide by at least 75% (Figure 4A), and at levels that can be achieved clinically (Figure 4B). The BRAF inhibitor vemurafenib did not influence MATE1 function, and this agent is presumably contraindicated because of its intrinsic nephrotoxic properties [45,46], which can cause potential delayed elimination of dofetilide independently of MATE1.

We next performed in vivo studies to assess if pretreatment with these contraindicated drugs elevates the plasma concentrations of dofetilide in mice in an OCT1/2- or MATE1-dependent manner. In male wild-type male pretreated with cimetidine or ketoconazole, representative inhibitors of the renal cation transport system and hepatic metabolism, respectively, the exposure to dofetilide was increased ~three-fold (Figure 4C–E). While plasma concentrations of dofetilide were slightly higher in female wild-type mice than in male mice (Figure 3E,F, Appendix A), which is consistent with clinical data [10], pretreatment of female mice with cimetidine or ketoconazole was associated with similar increases in exposure to dofetilide. These pharmacokinetic changes also occurred in OCT1/2^−/−^ mice and MATE1^−/−^ mice and could be replicated with other MATE1 inhibitors such as bictegravir, suggesting that the interaction can occur both at the level of basolateral and apical transport (Appendix A). Interestingly, the interaction liability for dofetilide with verapamil was sexually dimorphic (Appendix A). Although hormone-dependent regulatory mechanisms have been reported for renal organic cation transporters [47,48], and MATE1 expression is higher in the kidney of male mice (Appendix A), the clinical ramification of this sex-dependent influence of verapamil on the pharmacokinetics of dofetilide remains unclear.

### 2.5. Influence of CYP3A on the Disposition of Dofetilide

Since dofetilide is at least partially metabolized by CYP3A4, it has been suggested that concurrent administration of dofetilide with inhibitors of this enzyme could potentially result in an increase in the plasma concentrations of Dofetilide [24,49,50]. To explore the likelihood of this interaction mechanism, we next performed pharmacokinetic studies with dofetilide in mice lacking all CYP3A isoforms (CYP3A^−/−^ mice) [51]. These studies revealed that the deficiency of CYP3A did not significantly influence the concentrations of dofetilide in plasma (Figure 5A,B,D,G) or urine (Figure 5C) after oral or i.v. drug administration. Furthermore, pharmacological inhibition of CYP3A with the prototypical inhibitor ketoconazole was associated with increased exposure to dofetilide in both wild-type mice and CYP3A^−/−^ mice (Figure 5A,D,E). These results support the thesis that clinical interactions of dofetilide with ketoconazole are unlikely related to the inhibition of hepatic metabolism but rather mechanistically connected with the modulation of renal OCT1/2- and/or MATE1-mediated transport.

### 2.6. Predicting the Interaction Liability for Dofetilide in Humans

To provide preliminary evidence for the translational significance of our murine data, we next applied a physiologically-based pharmacokinetic (PBPK) modeling approach to quantitatively predict drug effects in humans [52,53,54]. PBPK models utilize drug-dependent physicochemical and pharmacokinetic parameters along with drug-independent physiological systems parameters [55,56], and while such integrated mechanistic strategies have been advocated by various regulatory agencies [57,58], PBPK models that would allow *a priori* prediction of transporter-mediated interactions with dofetilide have not been previously reported. Since PBPK models can predict experimentally unverified interactions and can provide dose adjustments in special populations [59], our primary objective was to develop a PBPK model for predicting transporter-mediated clinical interactions with dofetilide using a top-down approach incorporating in vitro and clinical data (Figure 6 and Appendix A, Appendix A). The developed PBPK model could adequately reproduce the observed plasma concentration-time profile and renal clearance after oral or i.v. drug administration (Figure 7A,B), and the simulated profiles corresponded well with experimental human data [60] (Appendix A).

Next, we applied the model to predict transporter-mediated interactions of dofetilide with cimetidine and ketoconazole reported previously in human subjects [61,62], and found an acceptable degree of concordance between the simulated and observed data (Figure 7C,D, Appendix A). This suggests that the PBPK model could be applied in the future to predict the influence of previously untested MATE1 inhibitors, such as certain tyrosine kinase inhibitors [63], on the pharmacokinetics of dofetilide in humans.

## 3. Materials and Methods

### 3.1. Chemical and Reagents

Parental human embryonic kidney (HEK293), and Chinese hamster ovary (CHO) cells were obtained from American Type Culture Collection (ATCC; Manassas, VA, USA). The cDNAs for the mouse and human plasmids of OCT1, OCT2, OCT3, OCTN1, OAT1, OAT3, MATE1, or MATE2-K were obtained from Origene (Rockville, MD, USA), and the reconstructed cDNAs were subcloned into an empty vector containing pcDNA5/FRT. The vector was transfected into HEK293 cells using the Flp-In system (Invitrogen, Waltham, MA, USA) and selected for expression using geneticin (G418). Cells were cultured in DMEM supplemented with 10% FBS and grown in a humidified incubator containing 5% CO_2_ at 37 °C. Contraindicated drugs of dofetilide were obtained from Sigma-Aldrich (St. Louis, MO, USA) or Selleckchem (Houston, TX, USA). Radiolabeled dofetilide was obtained from American Radiolabeled Chemicals (St. Louis, MO, USA).

### 3.2. Cellular Accumulation

Uptake experiments in human AC-16 cardiomyocytes, HEK293 cells overexpressing OCT1, OCT2, OCT3, OCTN1, MATE1, or MATE2-K, and CHO cells overexpressing OAT1 and OAT3 were performed with [^3^H] dofetilide using standard methods [64,65]. The cell culture and uptake conditions for cells expressing mMATE1 or hMATE1 were described previously [63,66]. All results were normalized to uptake values in cells transfected with an empty vector or DMSO-treated groups. In brief, cells were seeded in 12- or 24-well plates in phenol red-free DMEM containing 10% FBS, and were incubated at 37 °C for 24 h. After removal of the culture medium and rinsing with PBS, cells were preincubated with either DMSO or inhibitors for 15 min followed by the addition of indicated substrate for 15 min. Uptake studies were performed with [^14^C] TEA (2 µM) or [^3^H] dofetilide (1 µM) in the presence or absence of inhibitors for a period of 15 min, unless stated otherwise. The uptake experiment was terminated by washing three times with ice-cold PBS. Cells were lysed in 1N NaOH at 4 °C overnight, and then the solution was neutralized with 2M HCl. Total protein was measured using a Pierce BCA Protein Assay Kit (Thermo Scientific, Waltham, MA, USA), and total protein content was quantified using a microplate spectrophotometer. Intracellular drug concentrations were determined in the remaining cell lysate by liquid scintillation counting.

The transcellular transport assay of [^3^H] dofetilide or [^14^C] metformin (1 µM) were performed in monolayers of single-transfected MDCK cells overexpressing human OCT2 or MATE1 and double-transfected human OCT2/MATE1, as previously described [37,67]. Transcellular transport was quantified by measuring the amount of dofetilide appearing in the apical compartment after 60 min of incubation.

### 3.3. Ex Vivo Cardiomyocytes Uptake

Cardiomyocytes from wild-type and MATE1-deficient mice were isolated as previously described [26,68], and used for ex vivo cardiomyocyte uptake assays with dofetilide. Briefly, hearts from wild-type and MATE1-deficient mice were quickly removed and perfused on a Langendorff’s apparatus at 37 °C. After a 5 min perfusion with Ca^2+^-free tyrode solution containing (in mm): 140 NaCl, 5.4 KCl 0.5 MgCl_2_, 10 Hepes and 5.6 glucose; pH 7.3, the perfusate was then switched to tyrode solution containing Liberase Blendzymes (Roche, Applied Science, IN) for digestion of the connective tissue. After 20 min of digestion, cardiomyocytes were isolated from dissected and triturated hearts and stabilized in BSA containing tyrode solution.

Isolated cardiomyocytes were then plated in 12 well plates containing Ca^2+^ and Mg^2+^ free Hank’s balanced salt solution. Cells were preincubated with either DMSO or cimetidine (25 µM) for 15 min followed by the addition of [^3^H] dofetilide (2 µM) for 30 min. The uptake experiment was terminated by washing three times with ice-cold PBS. Cells were lysed in 1N NaOH at 4 °C overnight, and then the solution was neutralized with 2M HCl. Total protein was measured using a Pierce BCA Protein Assay Kit (Thermo Scientific, Waltham, MA, USA). Intracellular drug concentrations of dofetilide were measured in the remaining cell lysate by liquid scintillation counting.

### 3.4. Gene Expression Analysis

RNA was isolated from adult or neonatal wild-type hearts and kidneys (30 mg) as well as from cardiomyocyte cells isolated from wild-type and MATE1-deficient mice. Tissues and cardiomyocytes were homogenized and then RNA was extracted using an EZNA Total RNA Kit extraction kit (Cat# R6834-02, Omega Bio-tek, Norcross, GA, USA). cDNA was generated from 2 μg of RNA using qScript XLT cDNA Supermix (Cat# 95161-100, Quantabio, Beverly, MA, USA). Real-time reverse transcriptase PCR (RT-PCR) was performed with TaqMan primer (Mm00840361_m1, Thermo Fisher Scientific, Waltham, MA, USA) and TaqMan Fast reagents. Reactions were carried out in triplicate, and normalized to Gapdh (Mm99999915_g1, Thermo Fisher Scientific, Waltham, MA, USA).

### 3.5. Protein Analysis

Isolated hearts from wild-type and MATE1-deficient mice were extracted and lysed using sonication. Pierce Bicinchoninic Acid (BCA) Protein Assay Kits (Thermo Fisher Scientific, Waltham, MA, USA) were used to determine protein concentrations. Next, an equal amount of protein was separated on a Bis-Tris 4–12% SDS-polyacrylamide gel with MOPS buffer according to the instructions from manufacturer (Life Technologies, Grand Island, NY, USA) and transferred to PVDF membranes. Western blot analysis was performed using antibodies against mouse MATE1 (Cat # 20898-1-AP) obtained from Proteintech Group, Inc (Rosemont, IL, USA), vinculin (Cat # 13901S), and HRP-conjugated secondary anti-rabbit (Cat # 7074) obtained from Cell Signaling Technology (Danvers, MA, USA). Proteins were visualized by chemiluminescence using the SignalFire ECL Reagent (Cell Signaling Technology, Danvers, MA, USA) or SuperSignal West Femto Maximum Sensitivity Substrate (Invitrogen, Carlsbad, CA, USA) using film.

### 3.6. Immunohistochemistry

Heart and kidneys were collected from mice and immersed in 10% neutral buffered formalin for 72 h at room temperature. Tissue sections of 4 μm were obtained from formalin-fixed, paraffin-embedded heart and kidney tissue blocks, and were mounted on Superfrost Plus glass slides. They were heated at 90 °C for 20 min, deparaffinized in xylene, and rehydrated with a series of graded ethanol. Antigen retrieval was performed by treating slides in 1X Tris-buffered saline (TBS) for 20 min at 90–95 °C. Endogenous peroxidase activity was quenched by incubating slides in 3% H_2_O_2_ for 15 min at room temperature. The sections were blocked with Avidin/Biotin blocking solution (Vector Laboratories, Burlingame, CA, USA, Cat # SP-2001) for 15 min each.

Primary antibody (mouse SLC47A1, Bioss Antibodies Inc., Woburn, MA, USA, Cat # BS-9284R) was used at a concentration of 2.5 µg/mL for 30 min, and secondary antibody (Donkey anti-rabbit, Jackson ImmunoResearch Inc., West Grove, PA, USA, Cat # 711-065-152) at 5 µg/mL for 30 min followed by adding a tertiary reagent, Vectastain^®^ Elite ABC-HRP Reagent (Vector Laboratories, Inc., Burlingame, CA, USA, Cat # PK-7100) for 30 min. Immunoreactive sides were detected using the DAB substrate kit (Agilent, Santa Clara, CA, USA, Cat # K3468) for 5 min. Slides were counterstained with Richard-Allan hematoxylin 2 (Fisher Scientific, Waltham, MA, USA, Cat # 7231). Slides were then processed with a sequential ascending alcohol series for dehydration and xylene series for clearing followed by mounting a coverslip.

### 3.7. In Vivo Electrocardiographic Recordings (ECG)

Continuous ECG recordings (PL3504 PowerLab 4/35, ADInstruments) were obtained from wild-type and MATE1-deficient neonatal mice (one day old) according to a previously described method [69,70]. Briefly, baseline ECG was recorded for 5 min, neonatal mice received a single i.p. dose of dofetilide (0.5 mg/kg) dissolved in sterile saline containing DMSO (20:1), and ECG recording continued for 20 min as described previously [31]. ECG recordings were analyzed using the LabChart 7.3 software (ADInstruments). The QT interval was measured before and after the administration of dofetilide from the beginning of the QRS complex to the isoelectric baseline for T waves [70]. Heart rate-corrected QT (QTc) intervals were then obtained using the formula QTc = QT/(RR/100)^1/2^ [71].

### 3.8. Animal Models

All animals were housed in a temperature-controlled environment with a 12-h light cycle, given standard diet and water *ad libitum*, and handled according to the Animal Care and Use Committee of The Ohio State University, under an approved protocol (2015A00000101-R2). All experiments were performed with male or female mice from wild-type FVB mice or with age- and sex-matched (8–15 weeks) mice engineered to be deficient in all CYP3A isoforms (CYP3A^−/−^), OCT1 and OCT2 (OCT1/2^−/−^), MATE1 (MATE1^−/−^), or OCT1/2 and MATE1 (OCT1/2/MATE1^−/−^). MATE1-deficient mice on a C57BL/6 background were obtained from Dr. Yan Shu (University of Maryland, Baltimore, MD, USA) and serially backcrossed to a pure FVB background. Previous studies have indicated that the genetic loss of MATE1 is associated with increases in the systemic exposure and accumulation in the heart, liver, and kidneys of cationic-type substrates [72,73]. OCT1/2/MATE1-deficient mice were generated by crossbreeding OCT1/2- and MATE1-heterozygotes. The deletion of OCT1/2 and MATE1 was verified at the level of DNA by PCR [74].

### 3.9. Pharmacokinetic Studies

For pharmacokinetic studies, plasma and tissue samples were collected from male or female wild-type mice (8–15 weeks old), and age-matched OCT1/2^−/−^, MATE^−/−^, and OCT1/2/MATE1^−/−^ mice following an established protocol [75]. Dofetilide was administered as a single oral (5 mg/kg) or i.v. (2.5 mg/kg) dose dissolved in sterile saline-1M HCl (399:1) with the pH adjusted to 7.4. The contraindicated drugs bictegravir (5 mg/mL), cimetidine (20 mg/mL), ketoconazole (10 mg/mL), trimethoprim (20 mg/mL), and verapamil (2 mg/mL) were dissolved in PEG400, and administered orally 30 min before dofetilide. Serial whole blood samples (0.083, 0.25, 0.5, 1, 3, and 6 h) were collected from the submandibular vein (3×), retro-orbital sinus vein (2×), or by cardiac puncture at the terminal time-point. Blood samples were centrifuged at 13,000 rpm for 5 min, and the plasma supernatants collected and stored at −80 °C until analysis. Dofetilide concentrations in heart were measured in wild-type and MATE1-deficient mice 15 min after a single i.v. dose of dofetilide (2.5 mg/kg).

To measure dofetilide concentration in urine, both male and female wild-type and age-matched OCT1/2^−/−^, MATE1^−/−^, and OCT1/2/MATE1^−/−^ mice were placed in Nalgene single mouse metabolic cages three days prior to the i.v. administration of dofetilide at a dose of 2.5 mg/kg. Animals had free access to a standard diet and water and were housed in a temperature- and light-controlled environment. Urine samples were collected in sterile 1.5 mL Eppendorf tubes at 24 h, 48 h, and 72 h post administration of dofetilide and stored at −80 °C until analysis. Plasma, urine, and tissue samples were analyzed by a validated method based on reversed-phase liquid chromatography coupled to tandem mass-spectrometric detection (LC-MS/MS) [76].

Pharmacokinetic parameters were calculated by non-compartmental analysis using Phoenix WinNonlin version 8.2 (Certara, New Jersey, NJ, USA). Peak plasma concentration (C_max_) was determined by visual inspection of the data from the concentration-time curves. The linear trapezoidal rule was used to obtain the area under the plasma concentration-time curve (AUC) over the sample collection interval. The relative heart exposure of dofetilide was calculated by determining the dofetilide concentration in the heart, corrected for contaminating blood, and dividing by the corresponding dofetilide concentration in plasma (ng/mL) at the 15 min time-point.

For TEA experiments, cimetidine (100 mg/kg) was given orally to wild-type and MATE1-deficient mice 30 min prior to the i.v. administration of [^14^C] TEA (0.2 mg/kg). Plasma samples were then collected after 15 min and measured for by liquid scintillation counting.

### 3.10. Physiologically Based Pharmacokinetic (PBPK) Modeling

#### 3.10.1. Input Parameters

PBPK models of dofetilide were developed using the software package SimCYP version 19 (Sheffield, UK). Simulations were carried out in the built-in healthy volunteer virtual population in the age range 20–50 years. Data from in vitro dofetilide uptake, inhibition of transport function, and observed pharmacokinetic and drug-drug interactions data in healthy subjects were used to develop the model (Appendix A). To simulate the effect of inhibitors (e.g., cimetidine or ketoconazole) on the pharmacokinetic profile of dofetilide, inhibitor models from the SimCYP drug library were directly used. The absorption of dofetilide was simulated using a first order absorption model with an absorption rate constant (k_a_) estimated from the plasma concentration-time profiles [60]. The volume of distribution at steady-state was also estimated using the plasma concentration-time profiles. The multicompartment mechanistic kidney model (EGD) was used to incorporate the glomerular filtration rate and renal tubular secretion of Dofetilide [77]. The latter was modeled measuring V_max_ and K_m_ values for OCT2 and MATE1 that were determined experimentally. The relative activity factor (RAF) value was set to 1 and 0.25 for OCT2 and MATE1, respectively.

#### 3.10.2. PBPK Modeling Strategy

The PBPKP model for dofetilide was developed following a top-down approach in which the multicompartment mechanistic kidney model was used to incorporate the glomerular filtration and active renal secretion of Dofetilide [77]. The modeling strategy employed in this project was based on the following considerations: (i) The renal tubular secretion of dofetilide was defined to be via OCT2 and MATE1 on the basolateral and apical membrane of the proximal tubular cells, respectively; (ii) The PBPK model was developed using in vitro and clinical data; (iii) The model was optimized by using publicly-available dofetilide plasma concentration-time profiles from healthy volunteers; (iv) The model was tested by using dofetilide dose given orally (0.5 mg, twice per day, BID), and i.v. infusion (0.5 mg) over 90 min, and the results were compared with clinical studies; and (v) The prediction of pharmacokinetic interactions between dofetilide and cimetidine or ketoconazole was carried out using the SimCYP drug library and results were compared with empirical data from clinical studies.

#### 3.10.3. Dofetilide Clinical Data

Plasma concentration-time profiles after oral administration of dofetilide at a dose of 0.5 mg, BID, and a 90-min i.v. infusion at a dose 0.5 mg in healthy volunteers were extracted from the published literature to build the model [60]. To evaluate the model predicted interaction with cimetidine and ketoconazole, study reports available from the US FDA clinical pharmacology and biopharmaceutics review were used [61].

#### 3.10.4. PBPK Simulations

The drug-drug interactions between dofetilide and inhibitors were assumed to occur at the level of the basolateral uptake transporter OCT2 and/or the apical efflux transporter MATE1. The K_i_ and f_u,inc_ values were obtained from in vitro experiments. In the simulations, cimetidine was administered orally at a dose of 400 mg BID for three days while ketoconazole was given orally at a dose of 400 mg QD for seven days. Inhibitors were administered from day one along with dofetilide through day three and day seven for cimetidine and ketoconazole, respectively. To evaluate the effect of the interaction between dofetilide and inhibitors, results were calculated and compared with observed and predicted AUC_0–last_ and C_max_ data.

### 3.11. Statistical Analyses

All data are presented as mean ± standard error of the mean (SEM), and experimental results from uptake studies were normalized to total protein content and baseline values, and expressed as a percentage. All experiments were performed using multiple replicates and were performed independently on at least two independent occasions. An unpaired two-sided Student’s *t*-test with Welch’s correction was used for comparisons between two groups (control/baseline vs. treatment/genotype), and a one-way ANOVA with Dunnett’s post-hoc test was used for comparing more than two groups. *p* < 0.05 was used as the statistical cut-off across all analyses.

## 4. Conclusions

We characterized a previously unrecognized transport mechanism of dofetilide mediated by MATE1 that regulates its efflux from cardiomyocytes and renal tubular cells. The function of this transport system is highly sensitive to pharmacological inhibition by a broad range of drugs and provides a mechanistic basis for previously reported pharmacokinetic interactions with dofetilide. These findings contribute to our understanding of the etiology of variable pharmacodynamic responses to initial dofetilide dosing and dofetilide-induced proarrhythmias. Furthermore, the findings suggest that caution is warranted when cardiotoxic MATE1 substrates are given together with inhibitors of this transport mechanism.

## Figures and Tables

**Figure 1 ijms-23-08607-f001:**
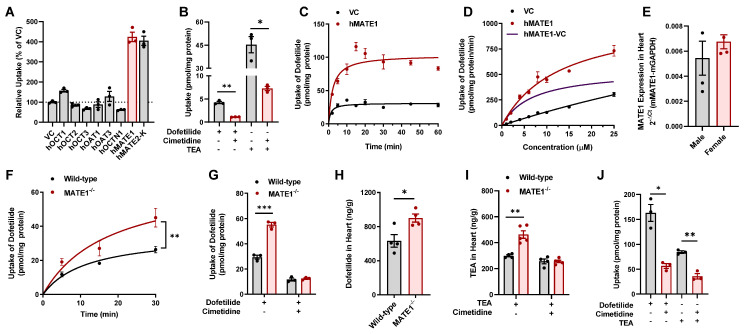
Inhibition of MATE1 enhances cardiac accumulation of dofetilide. (**A**) Transport of [^3^H] dofetilide (1 µM, 15 min) in cells overexpressing the human transporters OCT1, OCT2, OCT3, OAT1, OAT3, OCTN1, MATE1, or MATE2-K (2 min). Relative uptake is expressed as percentage change compared with empty vector controls (n = 3). (**B**) Relative uptake of [^3^H] dofetilide and [^14^C] TEA in HEK293 cells overexpressing human MATE1 in the presence and absence of cimetidine (25 µM). (**C**) Time dependent uptake (2–60 min) of [^3^H] dofetilide (1 µM) in HEK293 cells stably transfected with vector control (VC) or MATE1. (**D**) Transport kinetics of [^3^H] dofetilide in cells overexpressing human MATE1. The Michaelis-Menten constant (K_m_) and the maximal uptake rate (V_max_) values for the kinetics of dofetilide (1–25 µM) was determined after an incubation time of 2 min. K_m_ and V_max_ values for transport activity are 6.72 ± 1.71 µM, and 544.40 ± 55.70 pmol/min/mg, respectively. (**E**) Expression of the MATE1 gene in hearts isolated from untreated wild-type male and female mice (n = 4 per group). (**F**) Time dependent uptake of [^3^H] dofetilide (2 µM) in ex vivo cardiomyocytes isolated from wild-type or MATE1-deficient female mice (n = 4–6 per group). (**G**) Ex vivo concentrations of [^3^H] dofetilide (2 µM) in cardiomyocytes isolated from wild-type or MATE1-deficient female mice (n = 3 per group) for 30 min in the presence or absence of cimetidine (25 µM) pretreatment. (**H**) Concentration of dofetilide in whole heart tissue from wild-type or MATE1-deficient male mice 15 min after a single i.v. injection of dofetilide via the caudal vein at a dose of 2.5 mg/kg (n = 4 per group). (**I**) Concentration of TEA in whole heart tissue from wild-type and MATE1-deficient male mice (n = 4–5 per group) with or without treatment of with cimetidine (100 mg/kg) 30 min before an i.v. administration of [^14^C] TEA (0.2 mg/kg). Heart samples were collected 15 min after TEA administration. (**J**) Uptake of 2 µM [^3^H] dofetilide and [^14^C] TEA in AC16 human cardiomyocytes (n = 3) for 20 min in the presence or absence of cimetidine (25 µM) pretreatment (15 min). All experimental values are presented as mean ± SEM. Statistical analysis was performed using an unpaired two-sided Student’s *t*-test with Welch’s correction: * *p* < 0.05, ** *p* < 0.01, *** *p* < 0.001.

**Figure 2 ijms-23-08607-f002:**
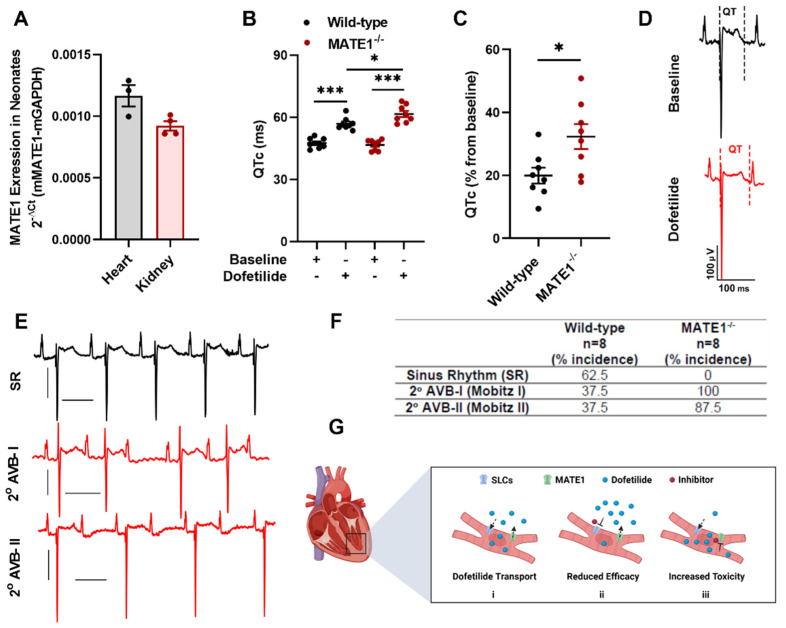
MATE1 deficiency exacerbates dofetilide-induced proarrhythmia. (**A**) Expression of the MATE1 gene in heart (n = 3) and kidneys (n = 4) isolated from untreated neonatal wild-type mice (day 1). (**B**) QTc interval and (**C**) percent QTc interval changes in neonatal wild-type or MATE1-deficient mice (day one) 15 min after a single i.p. injection of dofetilide at a dose of 0.5 mg/kg (n = 8 per group). All experimental values are presented as mean ± SEM. Statistical analysis was performed using an unpaired two-sided Student’s *t* test with Welch’s correction: * *p* < 0.05, *** *p* < 0.001, compared to baseline values. (**D**) In vivo surface ECG illustrating changes in QTc after dofetilide treatment in neonatal wild-type mice. (**E**) Dofetilide-induced second-degree AV blocks in neonatal MATE1-deficient mice. (**F**) Incidence of second-degree AV blocks (Mobitz I and Mobitz II) in neonatal wild-type and MATE1-deficient mice (n = 8 per group) after treatment with dofetilide. (**G**) Schematic diagram illustrating the proposed MATE1-dependent regulation of dofetilide transport (i) as a modulator of accumulation in cardiac myocytes leading to reduced (ii) or increased (iii) intracellular concentrations and reduced or increased electrophysiologic activity.

**Figure 3 ijms-23-08607-f003:**
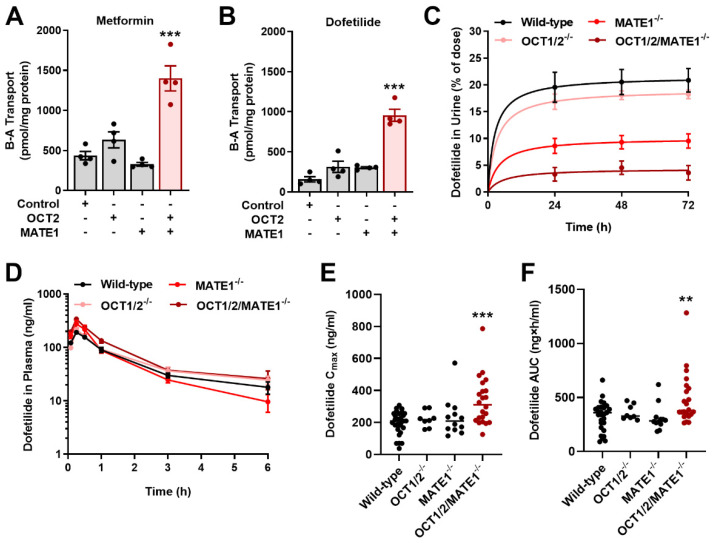
Inhibition of MATE1 reduces renal elimination of dofetilide. Characterization of the basolateral to apical (B-A) transport of [^14^C] metformin (**A**) [^3^H] dofetilide (**B**) in single transfected MDCK-VC, MDCK-OCT2, MDCK-MATE1, and double-transfected MDCK-OCT2-MATE1 cell lines. Transcellular transport was quantified by measuring the amount of metformin or dofetilide added basolaterally to the monolayers and appearing in the apical compartment after a 60-min incubation. Statistical analysis was performed using an unpaired two-sided Student’s *t*-test with Welch’s correction: *** *p* <  0.001 vs. MDCK-VC. (**C**) Urinary excretion of dofetilide in female wild-type, OCT1/2-deficient, MATE1-deficient, and OCT1/2/MATE1-deficient mice (n = 5) following a single i.v. dose of dofetilide (2.5 mg/kg). (**D**) Plasma concentration-time profile of dofetilide in female wild-type, OCT1/2-deficient, MATE1-deficient, and OCT1/2/MATE1-deficient mice (n = 5) receiving a single oral dose of dofetilide (5 mg/kg). (**E**) Peak concentration (C_max_) of dofetilide and (**F**) area under the curve (AUC) of dofetilide after a single oral dose in female mice of varying transporter genotypes. Statistical analysis was performed using one-way ANOVA with Dunnett’s post hoc test: ** *p* < 0.01, *** *p* < 0.001, compared with wild-type mice. All data represent the mean ± SEM.

**Figure 4 ijms-23-08607-f004:**
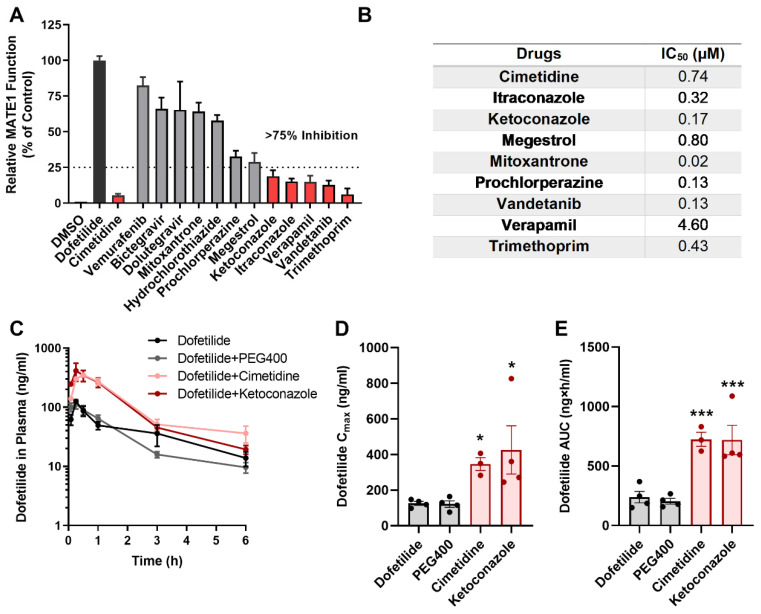
Drugs contraindicated for combined use with dofetilide inhibit MATE1 function. (**A**) FDA-listed contraindicated drugs of dofetilide were assessed at a concentration of 25 µM in HEK293 cells overexpressing human MATE1. [^3^H] Dofetilide (1 µM) and cimetidine were used as positive control substrate or inhibitor, respectively. Data are represented as the percentage residual MATE1 activity as compared with the vehicle control (DMSO) group (n = 3 per group). (**B**) IC_50_ values of different contraindicated drugs. (**C**) Plasma concentration-time curves profile of dofetilide in male wild-type mice receiving vehicle (PEG400), cimetidine (100 mg/kg), or ketoconazole (50 mg/kg) 30 min before dofetilide (n = 5 per group). (**D**,**E**) Pharmacokinetic parameters of dofetilide in male wild-type mice in the presence or absence of pretreatment with vehicle or contraindicated drugs. All data represent the mean ± SEM. Statistical analysis was performed using one-way ANOVA with Dunnett’s post hoc test: * *p* < 0.05, *** *p* < 0.001, compared with dofetilide alone control group.

**Figure 5 ijms-23-08607-f005:**
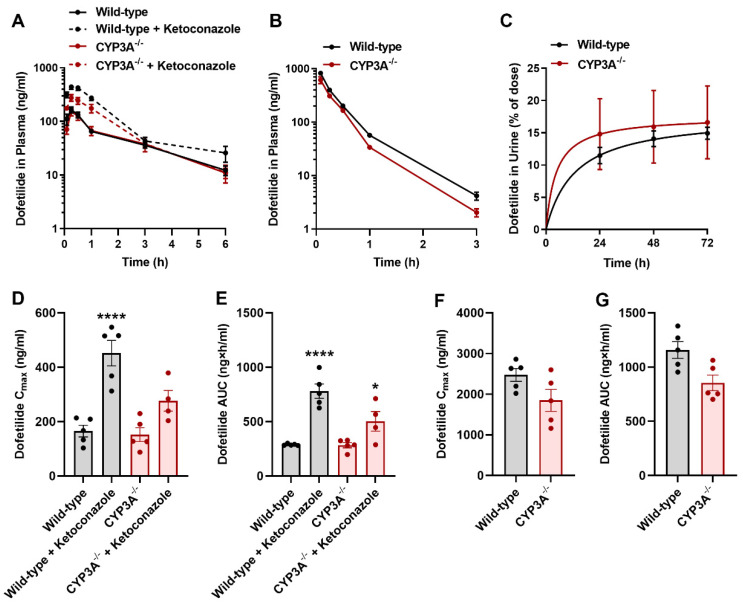
CYP3A inhibition does not influence the pharmacokinetics of dofetilide. (**A**) Plasma concentration-time profile of dofetilide (5 mg/kg, p.o.) in male wild-type or CYP3A-deficient mice pretreated with ketoconazole (100 mg/kg) 30 min before dofetilide (n = 5 per group). (**B**) Plasma concentration-time curves profile of dofetilide receiving an i.v. dose of 2.5 mg/kg in male wild-type or CYP3A-deficient mice (n = 5). (**C**) Urinary excretion of dofetilide in male wild-type and CYP3A-deficient mice (n = 5) following a single dose of dofetilide (2.5 mg/kg, i.v.). Pharmacokinetic parameters of dofetilide in male wild-type and CYP3A-deficient mice receiving an oral (5 mg/kg) (**D**,**E**), and i.v. (2.5 mg/kg) dose (**F**,**G**). Statistical analysis was performed using one-way ANOVA with Dunnett’s post hoc test: * *p* < 0.05, **** *p* < 0.0001 compared with wild-type mice receiving vehicle alone. All data represent the mean ± SEM.

**Figure 6 ijms-23-08607-f006:**
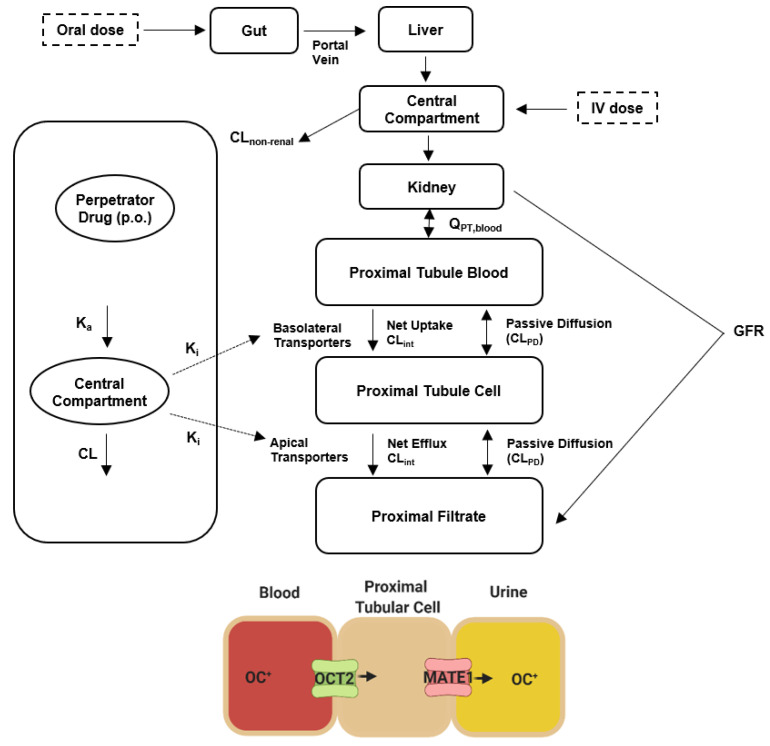
Structure of the physiologically based pharmacokinetic (PBPK) model for dofetilide. Abbreviations: CL, clearance; GFR, glomerular filtration rate; OC^+^, organic cation; K_a_, absorption rate constant; K_i_, inhibition constant; Q, blood flow.

**Figure 7 ijms-23-08607-f007:**
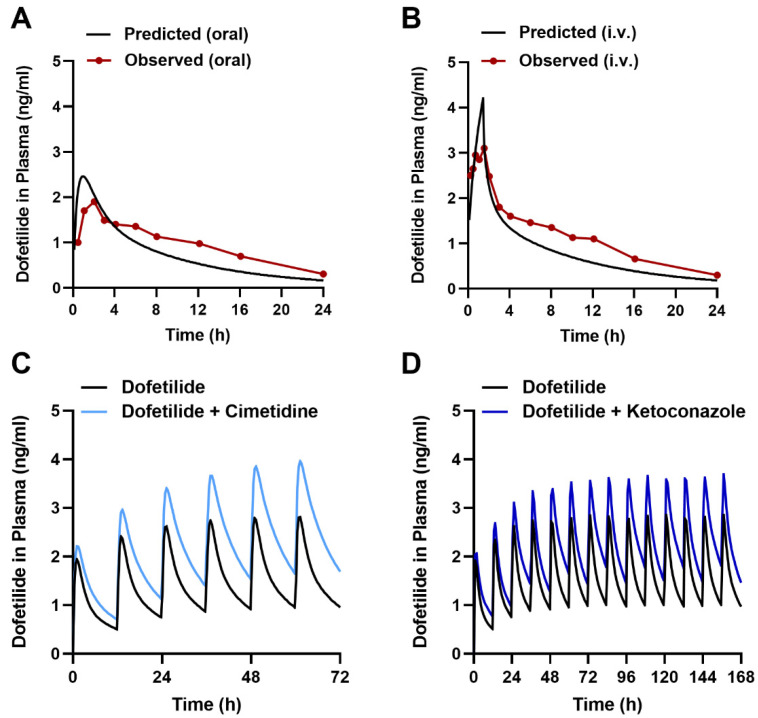
PBPK modeling predicts transporter-mediated interactions with dofetilide in humans. Observed and predicted plasma concentration-time profile after a single dose of 0.5 mg dofetilide oral administration (**A**), and 90-min i.v. infusion (**B**) at a single dose of 0.5 mg in adult healthy volunteers. Predicted plasma concentration-time profiles receiving multiple oral doses of dofetilide (0.5 mg, BID) with and without cimetidine (400 mg, BID) (**C**) and ketoconazole (400 mg, QD) (**D**) in healthy volunteers.

## Data Availability

The data that support the findings of this study are available from the corresponding author upon reasonable request.

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
