# Peer review of "MATE1 Deficiency Exacerbates Dofetilide-Induced Proarrhythmia"

_ijms, 2022, doi:10.3390/ijms23158607_

Round 1

Reviewer 1 Report

The manuscript by Uddin and his colleagues identified MATE1 as a transporter of dofetilide, characterized the transport mechanism of dofetilide mediated by MATE1, and built a physiologically-based pharmacokinetic model to predict the drug-drug interactions. This paper is interesting, and fits the scope of the Special Issue “Overcoming Biological Barriers: Importance of Membrane Transporters in Homeostasis, Disease, and Disease Treatment”. It would be advisable to briefly present the known mechanisms of other AADs mediated by transporters in hearts in the introduction part if there are any. I also found Figure S4 eye-catching and would like to move it to the main text. There are some minor spelling and style errors, such as line 436 “rabit”(rabbit) or “Journal of the American Heart Association” (J. Am. Heart Assoc.) at line 601 that the authors should check.

Author Response

Dear Reviewer,

Thank you so much! We appreciate your thoughtful comments. Please see below the edits we made in our main text as suggested:

a) We have added a brief statement in the introduction mentioning the known transport mechanism of some anti-arrhythmic drugs facilitated by organic cation transporters in the heart.

New sentence (line 74-76):

“Previous studies have reported the involvement of several organic cation transporters facilitating the transport of antiarrhythmic drugs [13,14], and potentiating the blockage of HERG potassium channel [15].”

b) We have moved “Figure S4” into the main text represented as “Figure 7” in lines 339, and 359.

c) We have corrected the spelling in “line 436”, and changed the journal abbreviation in “line 601” accordingly.

Reviewer 2 Report

Dear Authors,

I read the article submitted for my review with great interest. The manuscript is written very carefully, and all analyzes are in-depth.

The title is correct and reflects the content of the article.
The Introduction is an excellent presentation of the subject based on the current literature. The only thing I would add here is more information about multidrug and toxin extrusion 1 (MATE1);. However, they are woven into the Results and Discussion section; I think something in the Introduction could be added.

I found some confusion in figure 2. It concerns point G. There is no information about point A in the description. Besides, I recommend using different markings, as figure 2 already has A-C markings. Perhaps it would make sense to remove point G from here and insert it into a new figure, which should be better described in the present version; apart from the details mentioned above, it is unreadable.

Materials and Methods section

I didn't notice the description of the statistical analysis and what p was considered statistically significant?
This information is placed below the figures and tables, but I think if there was a separate section on generally used tests for analyzes, it would be helpful.
Apart from that, I have no comments; in my opinion, everything is consistent and correctly defined. The figures are precise; you can see everything clearly.

The conclusions are correct and are based on the results obtained.

Best regards

Author Response

Dear Reviewer,

Thank you so much for your thoughtful comments! We really appreciate your feedback. Please see below our edits based on your comments:

a) We have added more information about MATE1 transporter in the introduction.

New sentences (line 80-87):

“Multidrug and toxin extrusion protein (MATE1, SLC47A1) is predominantly expressed in the brush border membrane of renal proximal tubular cells and canalicular membrane of hepatocytes [16,17]. In addition, MATE1 is expressed at detectable levels in the heart [16,18]. Several prior studies have shown that MATE1 plays an important role in the distribution and elimination of endogenous and exogenous compounds [16,19], and functional variation or polymorphisms in MATE1 can affect the renal clearance of such compounds, including metformin and cimetidine [20–22]. Additionally, evidence suggests that inhibition of MATE1 could increase the area under the plasma concentration-time curve (AUC) of certain drugs [23].”

b) We apologize for this confusion. We have added a new marking style in Figure 2G for better readability. Existing marking A, B, and C in Figure 2G are replaced with i, ii, and iii.

New figure legend (Figure 2G) [line 199-202]:

(G) Schematic diagram illustrating the proposed MATE1-dependent regulation of dofetilide transport (i) as a modulator of accumulation in cardiac myocytes leading to reduced (ii) or increased (iii) intracellular concentrations and reduced or increased electrophysiologic activity.

c) Thanks for your suggestion. We have added a descriptive paragraph for statistical analysis. In our analysis, the p-value was set <0.05 to be considered as statistically significant.

New statistical analysis description (line 566-574):

“3.11. Statistical Analyses

              All data are presented as mean ± standard error of the mean (SEM), and experimental results from uptake studies were normalized to total protein content and baseline values, and expressed as a percentage. All experiments were performed using multiple replicates and were performed independently on at least two independent occasions. An unpaired two-sided Student’s t-test with Welch’s correction was used for comparisons between two groups (control/baseline vs. treatment/genotype), and a one-way ANOVA with Dunnett’s post-hoc test was used for comparing more than 2 groups. p < 0.05 was used as the statistical cut-off across all analyses.”